

# Investigating the effects of management practice on mammalian co-occurrence along the West Coast of South Africa

Deborah Jean Winterton[1,2], Nicola J. van Wilgen[1,3] and Jan A. Venter[2,4]

[1] Cape Research Centre, SANParks Scientific Services, Cape Town, South Africa
[2] School of Natural Resource Management, Faculty of Science, Nelson Mandela University, George, South Africa
[3] Centre for Invasion Biology, Stellenbosch University, Stellenbosch, South Africa
[4] Eugène Marais Chair of Wildlife Management, Mammal Research Institute, University of Pretoria, Pretoria, South Africa

## ABSTRACT

The subtle and cascading effects (e.g., altered interspecific interactions) that anthropogenic stressors have on local ecological assemblages often go unnoticed but are concerning given their importance in ecosystem function. For example, elimination of buffalo from the Serengeti National Park is suggested to have driven increased abundance of smaller antelope as a result of release from competition. The perceived low abundance of small antelope in the contractual Postberg section of the West Coast National Park (the park) has been an ongoing management concern which has been anecdotally attributed to predation by a mesopredator (the caracal, *Caracal caracal*). However, we hypothesized that the historical overstocking, and consequent overgrazing by larger-bodied managed ungulates would influence small antelope abundance. Using camera traps, we investigated species co-occurrence and temporal activity between small antelope, managed ungulates and caracals in Postberg as well as another part of the park (Langebaan) and a farm outside of the park. Results suggest that small antelope and managed ungulates have a high degree of temporal overlap ($\Delta = 0.74$, 0.79 and 0.86 for the farm, Langebaan and Postberg respectively), while temporal partitioning between small antelope and caracal is apparent ($\Delta = 0.59$). Further, small antelope and managed ungulates appear to occur independently of one another (SIF = 0.91–1 across areas). Managed ungulates were detected almost three times more frequently on fallow lands when compared to the more vegetated sites within the park suggesting that segregated food/cover resources allow for independent occurrence. Small antelope had a much higher probability of occurrence outside of the protected area (e.g., $\psi = 0.192$ and 0.486 for steenbok at Postberg, Langebaan compared to 0.841 on the farm), likely due to less variable (more intact) habitat outside of the protected area. There is not sufficient evidence to currently warrant management intervention for predators. The small size of the protected area provides limited scope for spatial replication thus reducing possibilities to infer the cause and effect for complex interactions (which would historically have taken place over much larger areas) with negative implications for adaptive management. We recommend continued monitoring over multiple seasons and a wider area to determine the spatial information requirements to inform management of small protected areas.

Corresponding author
Deborah Jean Winterton,
deborah.winterton@gmail.com

## INTRODUCTION

Land use for anthropogenic gain results in habitat conversion, degradation and fragmentation which have, along with climate change, altered the biodiversity and ecosystems of the earth (*Chapin et al., 2000*; *Newbold et al., 2015*). Importantly, human activities also impact top-down and bottom-up ecosystem processes at the landscape level (*Burgi, Ostlund & Mladenoff, 2017*). Substantial knowledge gaps related to the local effects of global environmental change drivers persist, particularly those that do not manifest in obvious ways (*Newbold et al., 2015*). It is the more subtle and cascading effects (e.g., altered interspecific interactions or behaviour) that anthropogenic stressors have on local ecological assemblages that are particularly concerning given their importance in ecosystem function (*Erb et al., 2017*; *Frey et al., 2017*). For example, the re-introduction of wolves to Yellowstone, which culminated in the regeneration of aspen trees, and altered river flow dynamics, produced insights into trophic cascades and ecosystems function driven by species interactions (*Berger & Conner, 2008*; *Beschta et al., 2016*; *Fortin et al., 2005*; *Newsome & Ripple, 2015*; *Ripple et al., 2014*). Niche partitioning between species at the same trophic level is an important facilitator of coexistence (*Frey et al., 2017*; *Herfindal et al., 2017*). Understanding how environmental stressors influence niche partitioning between species is critical for informing management decisions as well as for improving our understanding of how local assemblages respond to anthropogenic changes (*Frey et al., 2017*).

Since species have evolved traits in response to variation within their environment, the environment essentially determines species distribution and abundance, which shapes populations (*Molles, Cahill & Laursen, 1999*) and social organisation (*Jarman, 1974*). Resource partitioning is commonly observed in diet segregation, habitat use, spatial aggregation or timing of peak activity, which is needed for sympatric species to co-exist (*Herfindal et al., 2017*; *Marti et al., 1993*). Resource partitioning amongst ungulates is usually driven by body size (*Cromsigt & Olff, 2006*), competition and resource availability (*Gordon & Illius, 1989*). For example, smaller antelope have higher per-mass metabolic rates, thus requiring higher quality forage compared to larger ungulates, with lower relative metabolic rates, who rely on consuming larger quantities of forage (*Cromsigt & Olff, 2006*). As such, resources are partitioned along a niche axis of quantity versus quality (*Cromsigt & Olff, 2006*). Forage quantity varies vertically (accessible plant height), as well as horizontally (heterogeneity of vegetation and patch size), as such influencing behaviour (*Cromsigt & Olff, 2006*; *Venter et al., 2014*). Time can also be considered a resource (*Frey et al., 2017*), where time spent on one activity is a lost opportunity for gains of another activity (e.g., forage intake, predation risk and thermoregulation (*Owen-Smith & Goodall, 2014*; *Rowcliffe et al., 2014*)), which has associated energetic trade-offs. Animal activity level (time spent active) is, therefore, a good indicator of species energetics, foraging effort and
risk exposure, albeit poorly understood due to the challenges of quantifying activity in the field (*Rowcliffe et al., 2014*). Relative timing of species' activity levels may also be an indication of dominance (*Lazenby & Dickman, 2013*) or risk avoidance (*Díaz-Ruiz et al., 2016*; *Tambling et al., 2015*).

Land use and management practices affect habitat, with knock-on effects on species interactions, thus playing an important role in determining species distribution and abundance patterns (*Lazenby & Dickman, 2013*). Furthermore, direct alteration of species abundance and composition (e.g., through introduction, culling or species removal) could influence free-ranging species through facilitative or competitive interactions. A common example in Africa is farming of livestock in the presence of wild ungulates. While low abundance of domestic ungulates may improve foraging for wild ungulates (*Charles et al., 2017*), the effect becomes negative as densities increase (*Herfindal et al., 2017*). The majority of studies indicate that wild ungulates are negatively impacted by livestock, with the greatest negative responses being due to competition and a change in forage quantity and quality (see review by *Schieltz & Rubenstein, 2016*).

The apparent low abundance of small antelope (common duiker, *Sylvicapra grimmia*, and steenbok, *Raphicerus campestris*) in the contractual Postberg section of the West Coast National Park (hereafter referred to as the park) has been a management concern since the early 1990s (*Avenant, 1993*; *Heydenrych, 1995*). There is a perception that these two species have lower abundance in the contractual section of the park, compared to other sections, which landowners have attributed to predation by a mesopredator, the caracal (JJ Fouche, Postberg owners consortium, pers. comm., 2015). Caracals (*Caracal caracal*) are the largest predator in the system following the historical extirpation of apex predators such as lions, leopards and hyenas. The caracal's "release" from the top down competition that would have occurred in the presence of large predators prior to extirpation (*Cruz-Uribe & Schrire, 1991*) may have enhanced its impact on small antelope populations in the area. However, while the predation threat may be realistic, it is not the only process that could result in low abundance of small antelope. Historical land use and management practice within Postberg includes agriculture (livestock and crop cultivation) and more recently the overstocking of large ungulates which has resulted in extensive habitat degradation and potentially competition (*SANParks, 2013*).

Using camera traps, we aimed to assess co-occurrence and temporal activity overlap between small antelope and managed ungulates (competition; Table 1) and caracal (predation) with a view to establish how land management influences species interactions; as such informing the need for wildlife management (e.g., population supplementation or removals). Camera traps are especially useful tools for observing animal interactions as they provide 24 h surveillance that can record multiple species over space with time-of-detection (*Lazenby & Dickman, 2013*; *Rowcliffe et al., 2014*), which is important for assessing interactions between co-occurring species. Furthermore, camera traps are instrumental in estimating species distribution and abundances in relation to anthropogenic change and stressors (*Frey et al., 2017*) as they are very effective at detecting medium to large sized terrestrial mammals (*Reilly et al., 2017*). We used data from cameras in three areas with different management practices and managed ungulate abundances (two within

**Table 1  Classification of ungulates by weight (ordered alphabetically per class).** No managed ungulates were classified within the "Small" class, but all species in the medium and large classes constitute managed species.

| Small ungulates (<25 kg)[a] | Medium ungulates (26–200 kg) | Large ungulates (>200 kg) |
|---|---|---|
| Common duiker (all areas) | Bontebok (Postberg and Langebaan) | Blue wildebeest (Postberg) |
| Steenbok (all areas) | Impala (Farm) | Cape mountain zebra (Postberg) |
| | Nyala (Farm) | Cattle (Farm) |
| | Red hartebeest (Langebaan) | Eland (Postberg and Langebaan) |
| | Sheep (Farm) | Gemsbok (Postberg) |
| | Springbok (Postberg) | Kudu (Postberg) |

Notes.
[a]One detection of a Cape Grysbok at the farm and one detection of Grey Rhebuk at Langebaan were not included in analyses.

the park, Postberg and Langebaan, and one on a research livestock farm) to assess how the presence or absence of managed ungulates and caracals affects the occupancy of small antelope (steenbok and common duiker) and whether there is a difference in temporal activity patterns between the species groups. We expected the presence of managed ungulates to negatively influence occurrence probabilities of small antelope in Postberg due to direct competition for resources with, or habitat modification by, the more abundant larger managed ungulates (*Fritz et al., 2002*). We also predicted a greater temporal overlap between managed ungulates and small antelope in Postberg when compared to more natural areas within the park due to the higher abundance of managed ungulates, thus forcing the small antelope to spend more time looking for appropriate resources. To add scope to the comparisons and influence of managed ungulates on small antelope we also investigated co-occurrence and temporal overlap between managed ungulates on a research farm where livestock are restricted to fenced camps. Here, as with Postberg, we expected the presence of livestock to have a negative influence on the occupancy of small antelope as well as to have higher temporal activity overlap when compared to the larger more natural area in the park where ungulates are free ranging. Finally, we expected a significant temporal niche partitioning between small antelope and caracal, to lower their predation risk (*Tambling et al., 2015*).

## MATERIAL AND METHODS

### Study area

This study took place in the largely nutrient-poor Fynbos Biome (*Mucina & Rutherford, 2012*), along the west coast of South Africa, where annual average rainfall varies between 152 mm in the north and 265 mm in the south. The predominant vegetation type of the region is Strandveld which is dominated by sclerophyllous, broad-leaved shrubs that form communities of medium density to closed shrublands (*Mucina & Rutherford, 2012*).

We defined three study areas (scenarios) by their different management practices where the 'stocking rate' and species of managed ungulates at each scenario acted as a proxy for 'management practice'. Managed ungulates were defined as those species that require the intervention of people to manage the populations e.g., removals, introductions, feed supplementation and census. Ungulates were classified into different size classes

according to weight (Table 1), based on the natural segregation of weight ranges between the managed ungulates, where animals <25 kg were classified as small, between 26 and 200 kg were classified as medium and anything >201 kg was classified as a large. No managed ungulate was classified in the small class. Two scenarios were within a protected area (Postberg and Langebaan sections of the West Coast National Park) and the third was on a research farm (Lamberts Bay, hereafter referred to as the farm). The park is located approximately 100 km north-west of Cape Town, South Africa, was proclaimed in 1985 and since then it has expanded to its current size of approximately 47,000 ha (*SANParks, 2013*). The farm is located outside of the town Lamberts Bay which is approximately 100 km north of the park (Fig. 1). The farm was selected due to its being of comparable size (the majority of commercial farms along the west coast are used for crop production with little livestock and are therefore not of a comparable size to the two areas within the park) while still being within the bioregion and having comparable vegetation types.

### Postberg
Postberg has historically been characterised by intensive management of large and medium-sized herbivores in a small area (1,800 ha). The land was originally acquired in the early 1800s by a group of farmers and was used primarily for winter grazing, but the land was also ploughed. Postberg was proclaimed as a private nature reserve in the 1960s, after which it was contractually included into the park in 1987. Many indigenous and extra-limital large and medium-sized ungulate species were introduced to Postberg since the 1960s which resulted in overgrazing of the small, fenced area. Managed ungulates at this site were estimated to occur at $\pm$ 11.3 animals/km$^2$ (Dataset S2) and included bontebok (*Damaliscus pygargus pygargus*), Cape mountain zebra (*Equus zebra zebra*), red hartebeest (*Alcelaphus buselaphus caama*), eland (*Taurotragus oryx*), blue wildebeest (*Connochaetes taurinus*), kudu (*Tragelaphus strepsiceros*), springbok (*Antidorcas marsupialis*) and gemsbok (*Oryx gazelle*). Although there have been consistent removal efforts of extra-limital and other ungulates since Postberg's inclusion into the park, November 2016 (6 months prior to the study) saw a significant removal of these species from the area, resulting in slightly lower densities during this study. Due to slow rates of ungulate reproduction and vegetation recovery, the past overabundance of large herbivores in the Postberg section was nonetheless expected to have a legacy effect of potential disturbance on vegetation and small antelope that would extend through the duration of this research.

### Langebaan
In the Langebaan scenario, large herbivores occur at lower densities and are not confined to a small area. Historically this area was also used for agriculture, which included livestock and crop production and different portions were proclaimed as part of the park in 1989 and 1996. Managed ungulates within the Langebaan scenario were eland, bontebok, and red hartebeest and occurred at around 7.6 animals/km$^2$ (C Cowell, SANParks, 2015, unpublished data).

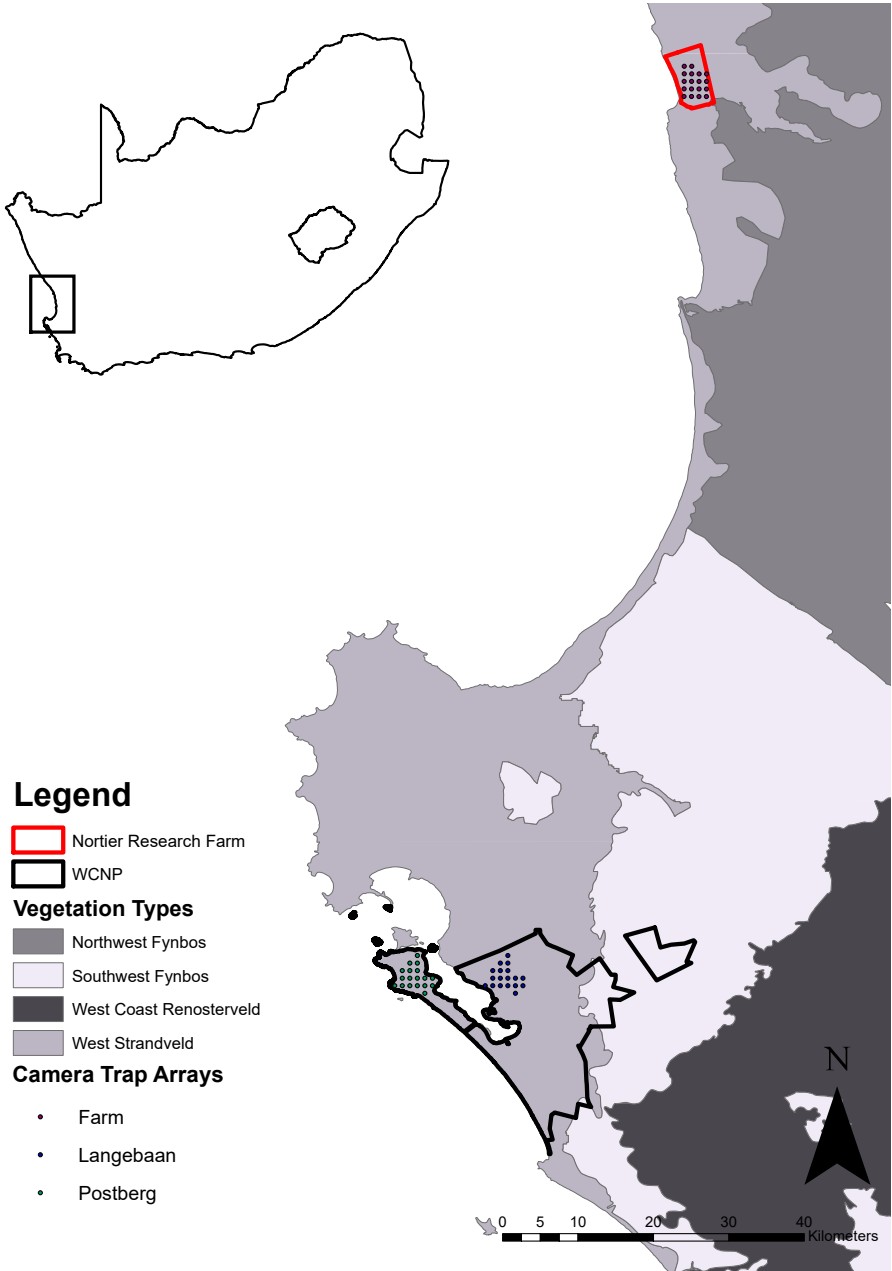

**Figure 1** **Map of the study area.** Vegetation type source: *South African National Biodiversity Institute (2006)*.

### Lamberts Bay (the farm)

The farm scenario made use of the Nortier research farm that falls under the management of the Department of Agriculture (Elsenburg), Western Cape Government and is 2,780 ha in size. Christie Rheeder, the manager of the farm, granted approval to conduct research at the farm on behalf of the Department of Agriculture (Elsenburg). Several resource flocks and herds are kept on the farm for the Directorate: Animal Sciences, while it is also the site

of veld rehabilitation projects run by the Directorate: Plant Sciences. The livestock present include sheep (*Ovis airies*), comprising three breeds (Namaqua Afrikaner, Dorper, and SA Mutton Merino), Bonsmara beef cattle (*Bos taurus*) and ostriches (*Struthio camelus*). The ostrich flock was restricted to camps that were not surveyed. In addition to the livestock, Impala (*Aepyceros melampus*) and Nyala (*Tragelaphus angasii*) are also present on the farm and were considered as managed ungulates, which were collectively estimated to occur at around 7.6 animals/km$^2$ (C Rheeder, research farm manager, pers. comm., 2018). The research farm is located in a matrix of other farms where predator control is known to take place.

## Survey design and in-field methods

The three scenarios were surveyed over the winter of 2017 using 18 Cuddeback®, model C3 blackflash (Non-Typical, De Pere, WI) camera traps. Postberg was surveyed between May 31st and July 3rd, followed by Langebaan between July 4th and August 7th and the farm from August 8th to October 16th 2017. We overlaid each area with a grid of 1 km$^2$ cells in ArcGIS (*ESRI, 2012*). The centroid of each of the 18 resultant cells served as the camera location. Once in the field, we used a handheld GPS to navigate to the centroid, after which we walked outward in a spiral fashion for up to 120 m from the centroid, seeking the first location where two or more signs of animal activity were detected (*Colyn, Radloff & O'Riain, 2018*). The camera was set at this point. Cameras were mounted approximately 40–50 cm above ground level, onto a wooden stake and faced in a southerly direction, away from the sun, to prevent false triggers and overexposure (*Glen et al., 2014*). The cameras were programmed to capture three burst photographs when triggered, with a 30-second delay between photographs.

Vegetation height and cover were measured at each site according to the protocol described in *Colyn (2016)*, i.e., by taking a measurement at one and two-meter distances from the camera trap location in a North, South, East and West direction (eight measurements in total). Vegetation height was recorded at the prescribed 1 m and 2 m distances with a measuring tape (short vegetation height allowed for this method) and percentage cover were also measured at these points using a densitometer (*Li, Peavey & Lane, 2000*). Height and cover were averaged across the eight measurements for use in analyses. Elevation was extracted at site level from the CGIAR-CSI SRTM 90 m Digital Elevation Data (*Jarvis et al., 2008*) using the spatial analyst tool within ArcGIS (*ESRI, 2012*). Similarly, slope was calculated using a digital slope model from the same DEM using the spatial analyst tool within ArcGIS (*ESRI, 2012*).

We predicted that the higher managed ungulate abundance at Posterg and the farm would result in increased competition between small antelope and managed ungulates. Further, we expected a lower overall vegetation height and abundance of small antelope at both Postberg and the farm due to the higher abundance of managed ungulates and we expected a lower abundance of mesopredators at the farm due to the surrounding predator suppression. Specific predictions in relation to each covariate are provided in Table 2.

**Table 2  Co-variates, data sources and predictions.** These data were used to model detection ($\rho$) and occupancy ($\psi$) per species. Effort was the only observation level co-variate considered.

| Co-variate | Variable type | Response variable | Source of data | Prediction |
|---|---|---|---|---|
| Management scenario | 3-level factor: Postberg, Langebaan, Lamberts Bay | $\rho$ & $\psi$ | Component of study design | Management is predicted to regulate managed ungulate abundance and affect vegetation height. Predictions in relation to particular species are made in the text. |
| Effort | Continuous variable, starting at 0 | $\rho$ | Camera traps | Increased effort is predicted to increase detection probability |
| Vegetation height | Continuous variable, starting at 0 | $\psi$ | Field data collection | Low vegetation height is predicted to have a negative effect on small ungulates due to cover requirements (*Heydenrych, 1995*) |
| Fallow land | Binary factor: 1—yes, 0—no | $\psi$ | Field data collection | Fallow lands are predicted to have a negative effect on small ungulates due to cover and high-quality forage requirements and a positive effect on managed ungulates due to availability of large quantities of albeit low-quality forage |
| Elevation | Continuous variable | $\psi$ | Digital Elevation Model (DEM) depicting elevation (m) at 90 m resolution (*Jarvis et al., 2008*) | Elevation range between sites and areas is 9–165 m above sea level and is expected to influence occupancy of all ungulates |
| Slope | Continuous variable | $\psi$ | Digital slope model depicting slope (°) at 90 m resolution (*Jarvis et al., 2008*) | Slope varies with topography and is expected to influence occupancy of all ungulates |
| Trail type | 2-level factor: road & game trail | $\rho$ | Field data collection | Preference for trail width and cover is expected to influence detection probability (*Mann, O'Riain & Parker, 2015*) |
| Large managed ungulate abundance | Continuous variable, starting at 0 | $\psi$ | Royle-Nichols Occupancy model output, based on data from camera traps | Areas of high large ungulate abundance were expected to decrease small ungulate occupancy due to competition and habitat modification |
| Medium managed ungulate abundance | Continuous variable, starting at 0 | $\psi$ | Royle-Nichols Occupancy model output, based on data from camera traps | Areas of high medium ungulate abundance were expected to decrease small ungulate occupancy due to competition and habitat modification |

## Data analysis

Camera trap images were downloaded and image data were processed and captured using the TimeLapse2 Image Analyzer software (Saul Greenberg, University of Calgary, Calgary, Alberta) and then exported to excel per camera station. Image databases of camera stations were merged for each management scenario. Non-animal photographs were removed from the database and when more than one species was captured in a single photograph, entries were duplicated and edited to capture the number of individuals of each species as separate records. If a photographed animal was not recognisable at species level but could be classified as either a small antelope or managed ungulate, then it was captured as such. Image data were binned into independent capture events using a loop in R (*R Development*

*Core Team, 2015*) which grouped all captures of a particular species, at a particular location. It then calculated the time difference between each picture and partitioned photographs of the same species at the same location into 30 min interval groups. The photograph with the highest number of individuals of that species was selected from each group and appended to the analysis database as an independent capture. We used the *tidyverse* and *ggplot2* packages (*Wickham, 2016*; *Wickham, 2017*) in R to manipulate databases and produce plots and summaries where necessary. Ungulate species diversity per area was calculated using the Shannon-Wiener index in the *vegan* package in R (*Oksanen et al., 2018*).

### Occupancy

Since Postberg had the shortest survey duration (32 days), the data from the farm and Langebaan were partitioned to use only the data collected in the first 32 days, which also ensured that only data collected over the wet season (31 May–10 September 2017), were used for the occupancy analyses. Postberg however also had one camera failure which resulted in 542 trap nights for Postberg, compared to 582 for Langebaan and 579 for the farm. To assess how the presence of managed ungulates influenced the occurrence of small antelope, we conducted single season, two-species occupancy analyses in PRESENCE (*Hines, 2006*) and ran single season, single-species occupancy models in R (*R Development Core Team, 2015*) using the *unmarked* package (*Fiske & Chandler, 2011*).

### Single-species occupancy

Site-specific covariates (Table 2) were captured in a separate site database and Pearson correlations assessed for those variables with numeric/continuous values. There was a strong positive correlation between vegetation height and cover ($r = 0.75$) and therefore we only used vegetation height in models as a proxy for vegetation structure.

We used the *camtrapR* package (*Niedballa et al., 2016*) in R to create a camera operation matrix and a detection history for individual species/or suites of species of interest (e.g., managed species). Temporal replication was defined per species by dividing the camera survey into sampling occasions which can range from 1 to 15 days (*Kok, 2016*). Occasion length varied for different species depending on their detectability. Shorter occasion lengths are better for assessing occupancy of species that are frequently detected, while longer occasion lengths are better for assessing occupancy of species that are less abundant and that have low detection probabilities. We experimented with different occasion lengths and settled on 5 days for caracal detection, 2 days for managed ungulate abundance and 7 days for steenbok and duiker occupancy.

Occupancy and abundance were analysed using the *unmarked* package in R (*R Development Core Team, 2015*) using the detection history of the species of interest, observation level, and site-specific covariates. Large and medium ungulate abundance per site was estimated using the abundance-induced heterogeneity model, *occuRN* function, (*Royle & Nichols, 2003*) in the *unmarked* package in R (*R Development Core Team, 2015*). Data were pooled for all three scenarios; effort was used to explain detection probability and scenario to explain occupancy. Small antelope occupancy and caracal detection were estimated using the single season, single species occupancy model implemented by the *occu* function (*MacKenzie et al., 2002*). Data across scenarios were pooled to estimate small

antelope occurrence and scenario was included to explain occupancy, while effort was included to explain detection for all species. The managed ungulate species' abundance estimates along with other site-specific covariates were used as predictors of small antelope occupancy. The best models were selected using the *modsel* and *fitlist* functions which produce a table of AIC and $R^2$ values. Models were assessed for relative goodness of fit by scrutinizing the AIC, delta and $R^2$ values in the model summaries (*Burnham, Anderson & Huyvaert, 2011*).

The "best" model is considered the model which produces the lowest AIC value. We also considered the delta value, which is the difference in AIC value between a model and the model with the lowest AIC. Earlier literature suggested that models with a delta value of >2 were poor, however recent evidence suggests that models with a delta value in the range of 2–7 should also be considered (*Burnham, Anderson & Huyvaert, 2011*). We made predictions based on the top models and inspected the 95% confidence intervals of the predictions (Table 3). Models that had no predictive power (i.e., produced lower and upper occupancy confidence estimates ranging between 0, i.e., complete absence, and 1, i.e., 100% occupancy) were not considered to be informative. If there were no covariates that provided strong predictive power, we made predictions based on either the null model or a model that used effort to explain detection and scenario to explain occupancy, depending on comparative AIC values.

Because many of the occupancy models did not have sufficient data or data variation to converge, we also tested particular hypotheses directly. We compared abundance, occupancy, vegetation height, detections and slope between each scenario. All these data were non-parametric (tested using the Shapiro–Wilk test of normality) and thus we used a Dunn's Test that accounts for tied ranks. We also compared variability of vegetation height between scenarios, using an asymptotic test in the *cvequality* package in R (*Marwick & Krishnamoorthy, 2018*). We hypothesised that managed ungulates would show a preference for fallow lands and as such would be detected more often, with a shorter time between captures than recorded elsewhere. Conversely, we expected small antelope to avoid these areas, and thus their time between captures on fallow lands would be higher (fewer detections). To assess this we calculated the number of detections for small, medium and large ungulates at each vegetated site and compared the number of detections on fallow versus natural lands across areas for an equal number of days (32). We also compared the average time between independent captures for species from each group (small/medium/large) between fallow and natural lands in Postberg and Langebaan. These analyses were not conducted for the farm as no fallow lands are present there.

### Temporal activity and overlap

Activity level and overlap of small antelope and managed ungulates were assessed using the *overlap* (*Ridout & Linkie, 2009*) and *activity* (*Rowcliffe, 2016*) packages in R (*R Development Core Team, 2015*). Prior to activity and overlap analyses, time was converted to decimal numbers. The time of sunrise and sunset was calculated using the *StreamMetabolism* package (*Sefick Jr, 2016*) and stored for each record based on the date and GPS location of the record. To increase the available sample size for the temporal activity and overlap

**Table 3  Top models for managed ungulate abundance and small antelope occupancy.**

|  | nPars | AIC | delta | AICwt | cumltvWt | Rsq |
|---|---|---|---|---|---|---|
| Large ungulate abundance |  |  |  |  |  |  |
| $\rho$ (effort), $\psi$ (scenario) | 5 | 614.62 | 0 | 1.0000 | 1 | 0.63 |
| $\rho$ (.), $\psi$ (.) | 2 | 661.02 | 46.4 | 0.0000 | 1 | 0 |
| Medium ungulate abundance |  |  |  |  |  |  |
| $\rho$ (effort), $\psi$ (scenario) | 5 | 488.95 | 0 | 0.99982 | 1 | 0.36 |
| $\rho$ (.), $\psi$ (.) | 2 | 506.23 | 17.29 | 0.00018 | 1 | 0 |
| Managed ungulate abundance |  |  |  |  |  |  |
| $\rho$ (effort), $\psi$ (scenario) | 5 | 898.47 | 0 | 0.9959 | 1 | 0.27 |
| $\rho$ (.), $\psi$ (.) | 2 | 909.43 | 10.96 | 0.0041 | 1 | 0 |
| Common duiker occupancy |  |  |  |  |  |  |
| * $\rho$ (effort), $\psi$ (scenario, veg height, elevation) | 7 | 288.4 | 0 | 0.2100 | 0.21 | 0.6 |
| * $\rho$ (effort), $\psi$ (scenario, veg height, medium ungulate abundance) | 7 | 289.49 | 1.08 | 0.1200 | 0.34 | 0.59 |
| * $\rho$ (effort, trail type), $\psi$ (scenario, veg height, elevation) | 8 | 290.38 | 1.98 | 0.0790 | 0.5 | 0.6 |
| * $\rho$ (effort), $\psi$ (scenario, slope, medium ungulate abundance) | 7 | 291.43 | 3.03 | 0.0470 | 0.54 | 0.57 |
| * $\rho$ (effort), $\psi$ (scenario, medium ungulate abundance) | 6 | 291.81 | 3.4 | 0.0390 | 0.63 | 0.55 |
| $\rho$ (effort), $\psi$ (scenario, veg height) | 6 | 292.48 | 4.08 | 0.0280 | 0.69 | 0.55 |
| Steenbok occupancy |  |  |  |  |  |  |
| * $\rho$ (effort), $\psi$ (scenario) | 5 | 236.15 | 0 | 0.0597 | 0.06 | 0.23 |
| $\rho$ (effort), $\psi$ (scenario, medium ungulate abundance) | 6 | 236.82 | 0.66 | 0.04284 | 0.1 | 0.25 |

**Notes.**

$\rho$, detection probability; $\psi$, probability of occurrence.

Co-variates used in the model are indicated in brackets while (.) indicates no co-variates were used. Modelled managed ungulate abundance outputs per site were used as co-variates in common duiker and steenbok occupancy models.

*These models showed no predictive power (confidence intervals were uninfomative, ranging between 0 and 1).

analyses, we lumped the data generated by the systematic survey described above with camera trap data that was opportunistically collected in the Postberg and Langebaan sections of the park between June 2016 and February 2017, where cameras were set on management tracks using the same camera settings as above as well as data generated from the full 60 days for which the farm was surveyed. This resulted in 1,150, 1,159 and 1,213 trap nights for the Postberg, Langebaan and the farm respectively. We did not have enough time-of-day observations for caracal at the farm and Postberg to assess temporal activity and overlap with small antelope, so this analysis was restricted to the Langebaan site.

The analysis of circular data is specialised, and standard statistical measures such as mean and variance, or regression are not appropriate (*Ridout & Linkie, 2009*). Time was converted to radians, a requirement of the *overlap* package. Activity was broadly depicted by non-parametrically estimating activity patterns using kernel density estimation with the bandwidth concentration parameter set at a maximum of 3 (*Ridout & Linkie, 2009*). This was further multiplied and adjusted by 1.5 as per *Rowcliffe et al. (2014)* who noted that bandwidth adjustment of 1.5 gave the most robust and minimally biased activity level estimations. The degree of overlap in temporal use of different species groups was estimated using the coefficient of overlap, $\Delta$, where 0 = complete separation and 1 =
complete overlap. If there were less than 75 observations for one of the species, the *Dhat1* overlap estimator was applied whereas if the sample size was greater than 75 for both species, *Dhat4* was used (*Ridout & Linkie, 2009*). We compared the activity overlap of the species of interest across scenarios first to assess whether they displayed any difference in activity patterns in the different areas. Following this, we compared the overlap between managed ungulates and small antelope at each scenario to determine if there is any evidence of temporal niche partitioning between species groups. Data were bootstrapped and resampled 500 times for each overlap estimate to generate 95% confidence intervals. To test whether activity patterns differed between species, we applied the Watson-Wheeler test of homogeneity for circular data to non-bootstrapped data using the *circular* package in R (*Agostinelli & Lund, 2017*; *Taşdan & Yeniay, 2014*). Activity levels and overlap were plotted using the *overlap* package.

## RESULTS

There was no significant difference in vegetation height between the sites ($p = 0.1627$), however, vegetation height on the farm was significantly less variable than in the park ($p = 0.002$). This was driven by the very high variation in vegetation height within Postberg and Langebaan, both of which included sites with no vegetation (former fallow lands) and well-vegetated sites with high vegetation height, whereas sites on the farm were more evenly vegetated.

### Species richness and abundance

Sixteen ungulate species were detected across the three areas during the study (see Table 1 for species; two species were not included in analyses because of single detections and/or detections in spring only). Highest ungulate species richness (nine) and diversity was recorded in Postberg (Shannon-Weiner diversity = 1.7), compared to Langebaan and the farm (Shannon-Weiner diversity = 1.15 and 1.14 respectively).

Abundance models for large ungulates had the best fit ($R^2 = 0.63$, Table 3). Model results reflected naïve occupancy (occupancy estimated without detection accounted for) results (naïve occupancy for large herbivores was 0.88, 0.941 and 0.05 for Postberg, Langebaan and the farm respectively, and 0.29, 0.33 and 0.72 for medium ungulates). Significant differences were detected in managed species abundance across sites. Medium sized ungulate abundance was significantly higher at the farm when compared to Postberg ($p < 0.0017$) and Langebaan ($p < 0.001$) with a mean site abundance of 1.62 individuals per site Fig. 2B. There was no difference in site abundance of managed medium-sized ungulates between Postberg and Langebaan ($p = 0.054$) with a mean of 0.511 and 0.371 individuals per site respectively. Large managed ungulates had a mean abundance of 0.074, 2.69, and 3.30 individuals per site for the farm, Langebaan, and Postberg respectively. As such, abundance was significantly lower at the farm compared to both park scenarios ($p = < 0.001$), while there was no difference between Langebaan and Postberg ($p = 0.19$; Fig. 2A). However, when large and medium-sized ungulates were pooled as 'managed ungulates' Postberg was estimated to have higher abundance than the farm ($p = 0.0461$) areas (Fig. 2C).

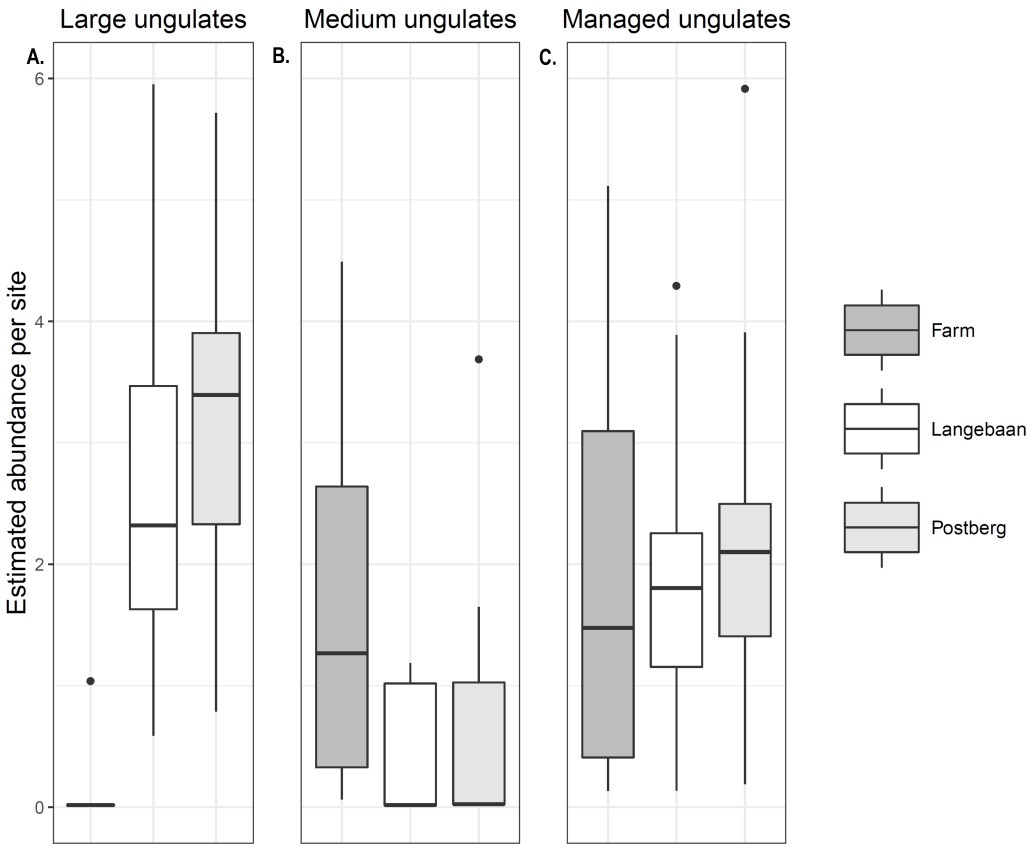

**Figure 2 Managed ungulate abundance.** Abundance estimated for (A) Large ungulates, (B) Medium ungulates and (C) Managed (large and medium) ungulates across the three scenarios.

## Detection probability

Differences in detection probability were also detected across areas and between the focal species groups ($p < 0.001$, Fig. 3). Caracal detection was highest at Langebaan ($\rho = 0.0520$), followed by Postberg ($\rho = 0.0398$) and was lowest at the farm ($\rho = 0.0253$). Common duiker, steenbok and small antelope (common duiker and steenbok lumped) detection differed significantly between all scenarios ($p < 0.001$). Detections were highest at the farm (0.778, 0.416, 0.867 respectively) followed by Langebaan ($\rho = 0.671, 0.384,$ 0.761 respectively) and Postberg ($\rho = 0.194, 0.0974, 0.256$ respectively). Conversely, large managed ungulates were detected most frequently at Postberg ($\rho = 0.275$) followed by Langebaan ($\rho = 0.202$) and were detected least frequently at the farm ($\rho = 0.105,$ $p < 0.001$). While large ungulates had the lowest detection probability at the farm, medium and managed (medium and large lumped) ungulates had higher detection probabilities there ($\rho = 0.324, 0.328$ respectively) compared to Postberg ($\rho = 0.261, 0.317$ respectively) and Langebaan ($\rho = 0.113, 0.231$ respectively).

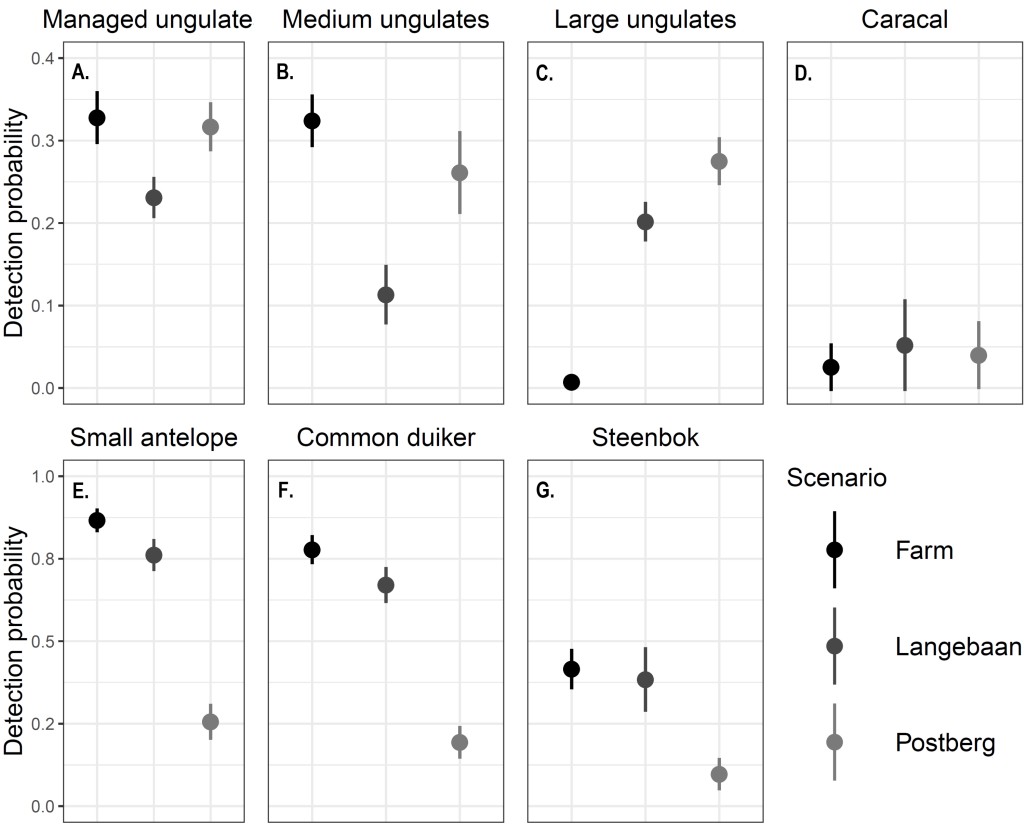

**Figure 3 Detection probability ± standard error of focal species.** The detection probability estimated for (A) Managed ungulates, (B) Medium ungulates, (C) Large ungulates, (D) Caracals, (E) Small Ante-lope, (F) Common duiker and (G) Steenbok across scenarios. Note that the scales differ between the first and second rows of graphs.

## Occupancy

We found that when detection probability was <0.1, occupancy could not be estimated, and therefore we could not make confident predictions for caracal occupancy for Postberg or the farm (Lamberts Bay). Estimates of naïve occupancy ranged from 0.11 at the farm to 0.16 and 0.17 at Langebaan and Postberg.

The results from the two species occupancy models were not particularly informative. Data were insufficient to run two-species occupancy models in Postberg, but models from the other scenarios suggest that ungulate species in different size classes occur independently of one another (see Supplemental Information).

Occupancy models for common duiker provided better goodness of fit compared to steenbok occupancy models ($R^2 = 0.55$ vs. 0.25, Table 3). Common duiker occupancy ($\psi$) differed significantly ($p < 0.001$) across scenarios with the farm having the highest occupancy of 1 ($\pm 0$ SE), followed by Langebaan ($\psi = 0.889 \pm 0.052$ SE) and Postberg ($\psi = 0.473 \pm 0.098$ SE). These estimates were very close to naïve occupancy: 1, 0.88 and 0.47 respectively (Fig. 4A). Steenbok occupancy was significantly different ($p < 0.001$) across all scenarios with the farm having the highest probability of occurrence ($\psi = 0.841 \pm 0.222$

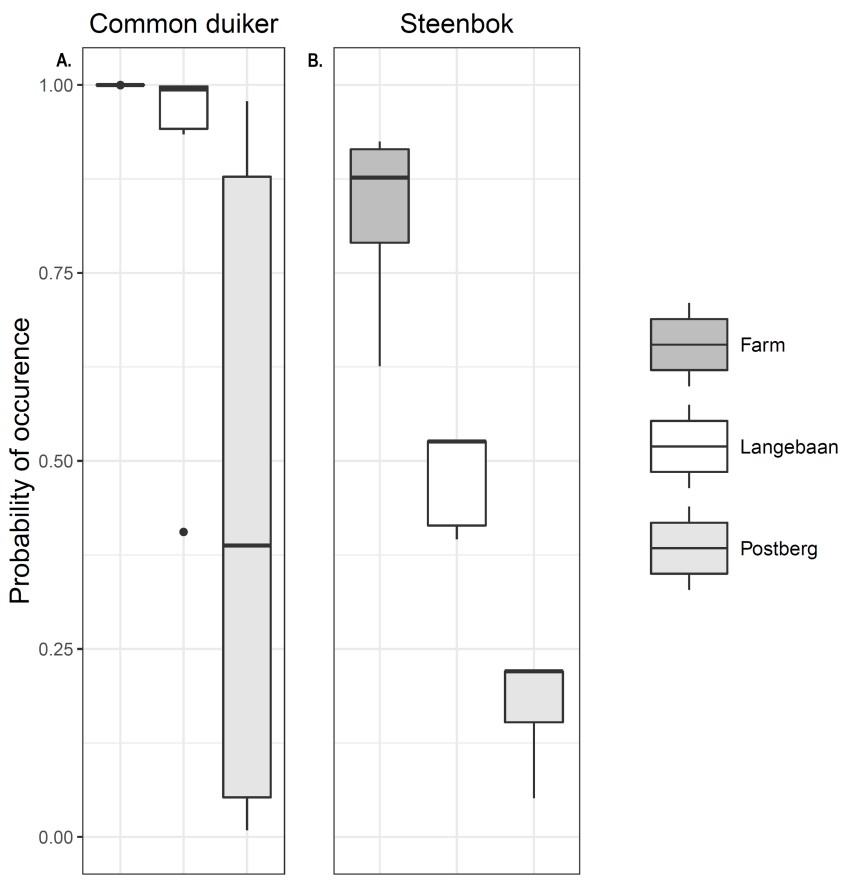

**Figure 4 Small antelope occurrence probability.** Probability of occurrence estimated for (A) common duiker and (B) steenbok across the scenarios.

SE; naïve $\psi = 0.77$) followed by Langebaan ($\psi = 0.486 \pm 0.013$ SE; naïve $\psi = 0.44$) and then Postberg ($\psi = 0.192 \pm 0.012$ SE; naïve $\psi = 0.17$) (Fig. 4B). Models for small antelope (common duiker and steenbok lumped) produced similar results to that of common duiker. Vegetation height was the strongest predictive variable for common duiker occurrence. While medium ungulate abundance produced the most reliable predictions for steenbok occurrence (Table 3), slope appeared to have a weak influence as it appeared in two of the top ten models. Estimates for the farm, however, remained uninformative due to 100% occupancy across sites.

Large managed ungulates were captured significantly more often on fallow lands ($p = 0.02$, Wilcox test), which were visited on average every 2.8 days, while natural lands were visited every 8.2 days across areas. This pattern was particularly prevalent in Langebaan, where most detections took place on or near fallow sites (Fig. 5A). Overall, there was no difference in small antelope detections between fallow and non-fallow lands however, the sample size of fallow sites was small and 32 detections at one fallow land site in Langebaan obscured potential patterns (Fig. 5A). There were no fallow land detections for small antelope at any of the five other fallow sites, inclusive of all fallow sites at Postberg.
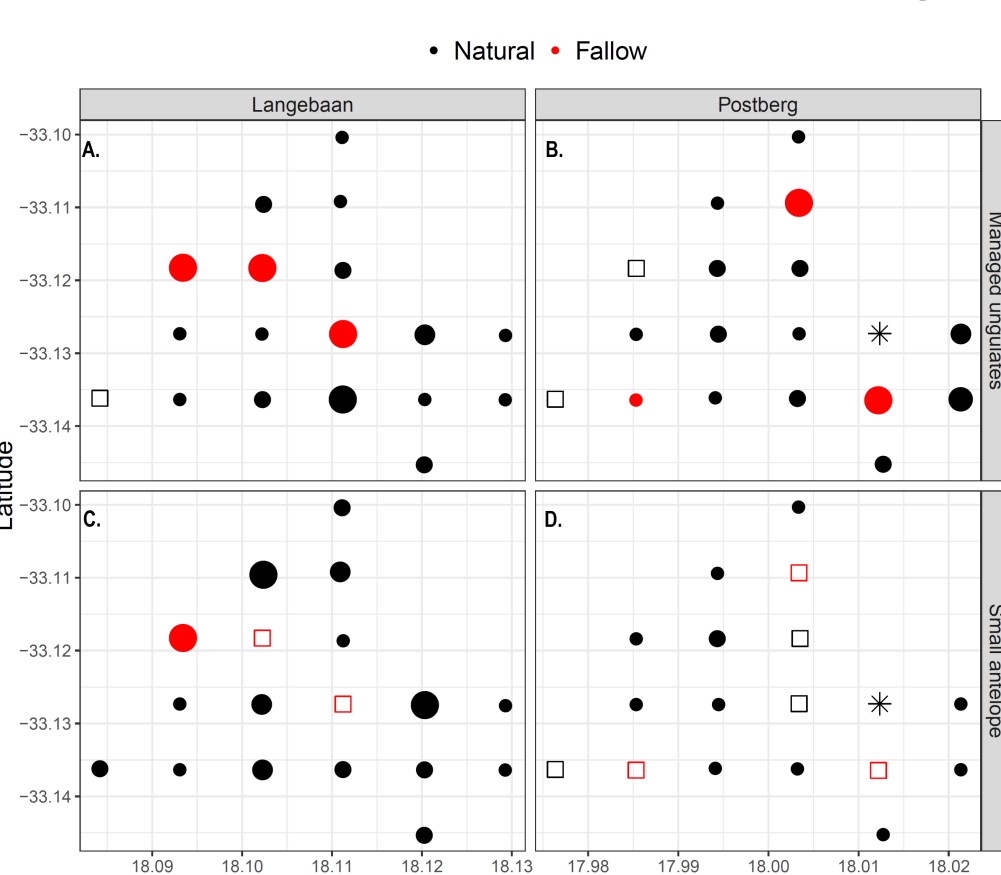

**Figure 5 Detection maps of managed ungulates (A–B) and small antelope (C–D) within the park.**
Points represent camera sites and the size and shape of the dots represent the frequency of detections at each site. Red dots/squares indicate fallow land sites.

## Temporal activity and overlap

Activity patterns were typical for the species of interest. Managed ungulates were primarily diurnal, small antelope crepuscular (Fig. 6) and caracals mostly nocturnal (Fig. S1). We assessed temporal overlap between caracal and small antelope independently as well as pooled (common duiker and steenbok) at the Langebaan site. The overall trend across the overlap analyses was for small antelope activity to peak at periods of low caracal activity (Fig. S1). Activity overlap between managed ungulates and small antelope was also assessed separately for each scenario (Fig. 6). As hypothesised, activity overlap was highest at Postberg (86%, CI [0.77–0.93]) followed by Langebaan (79%, CI [0.73–0.84]) and the farm (74%, CI [0.68–0.79]). At the latter two sites temporal activity was deemed to be significantly different ($p < 0.001$).

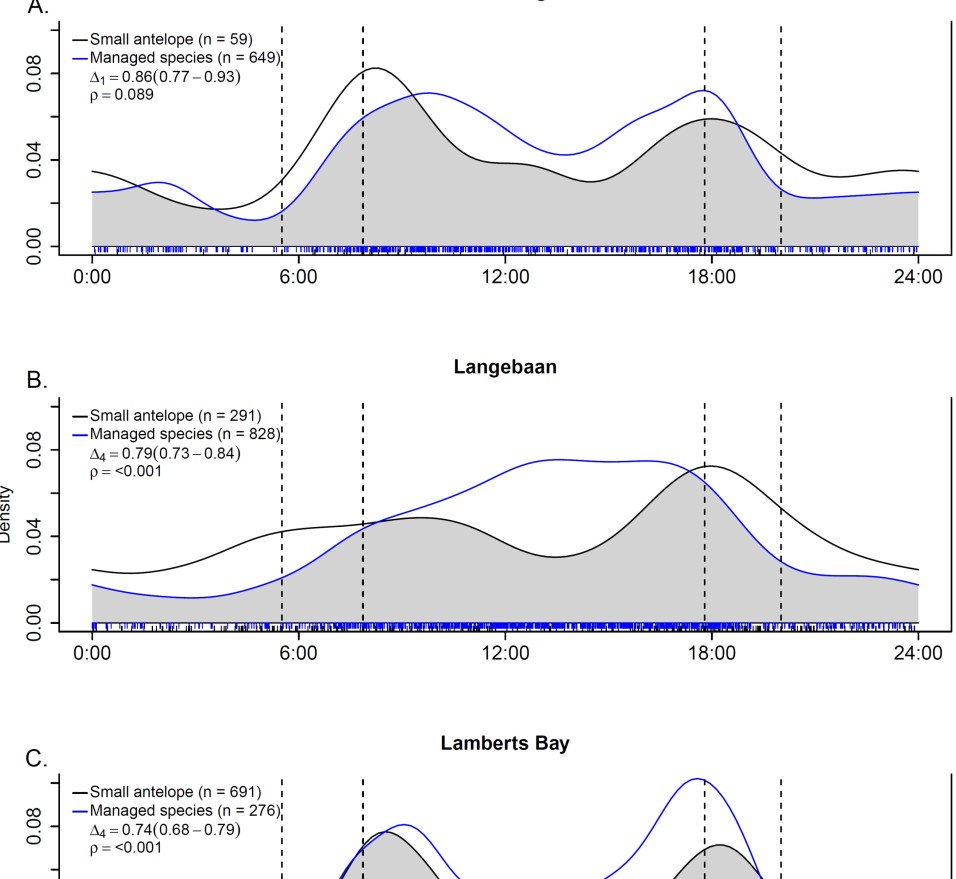

**Figure 6  Temporal overlap estimates between small antelope and managed ungulates at (A) Postberg, (B) Langebaan and (C) Lamberts Bay.** Time of day starts and ends at midnight on the *x*-axes and the fitted kernel-density is on *y*-axes. The grey shaded area indicates overlap and is described by the coefficient of overlap (Δ) and the associated estimator used (number in subscript) along with the 95% confidence intervals in parentheses. The vertical dotted lines represent the earliest and latest sunrise and sunset times across the study period. ρ is derived based on a Watson-Wheeler test of homogeneity for circular data. Dashed lines along the *x*-axes indicate the sample size of time-of-day observations.

# DISCUSSION

Co-existence between sympatric species requires segregation of resources such as food, habitat use, spatial distribution and temporal activity to facilitate niche partitioning (*Frey et al., 2017*; *Herfindal et al., 2017*). Adjusting temporal activity is also a way for prey species to avoid predation and escape risk (*Owen-Smith, 2015*). In this study, we explored the influence of different management practices on interspecific interactions by investigating

species co-occurrence and temporal activity overlap between small and managed ungulates as well as a potential predator, the caracal. Results suggest that small antelope and managed ungulates have a high degree of spatial and temporal overlap, while temporal partitioning between small antelope and caracal is apparent. A surprising finding was the much higher occurrence of small antelope outside of the park.

### Small antelope occurrence

As observed in other parts of Africa (*Caro et al., 1998*), management practice had an obvious impact on the occurrence of both steenbok and common duiker. Postberg consistently had the lowest probability of occurrence, with a much higher occupancy outside of the protected area at the farm (Fig. 4). Although few of our measured covariates had strong predictive power, vegetation structure was a good predictor for common duiker occurrence (Fig. S2), while medium ungulate abundance had a weak negative effect (Table 3) and slope a weak positive effect on steenbok occurrence (see results section; *Heydenrych, 1995*; *Arsenault & Owen-Smith, 2002*). This suggests that habitat and interactions with managed ungulates may be important drivers for the occurrence of small antelope within our study area. Therefore, we specifically explored interactions (i.e., competition and facilitation, habitat and predation) as possible mechanisms driving small antelope occurrence.

### Competition versus facilitation

The lower occurrence of small antelope at Postberg is unlikely to be due to medium ungulates since medium ungulate abundance was comparatively lower at Postberg than the farm. However, managed ungulates at the farm were primarily livestock restricted to camps, therefore, the potential influence on small antelope would be restricted within and not widespread across the area. We also considered the potential of managed ungulates facilitating forage opportunities for small antelope. However, facilitation is considered unlikely due to the nutrient-poor soils and slow growth rates of the vegetation which encourages competition between ungulates (*Fritz et al., 2002*). In addition, small antelope occurrence is lowest in the areas with the highest densities of managed ungulates, which may indicate competition (Fig. 5). For example, elimination of buffalo (*Syncerus caffer*) from the Serengeti National Park is suggested to have driven increased abundance of small antelope as a result of competition release (*Arsenault & Owen-Smith, 2002*). While there seems to be little interspecific competition at the local level, this might not be the case at a larger scale. For example, roe deer and wild boar have been seen to occur independently of cattle at the habitat level in China but displayed segregation at the landscape level (*Wang et al., 2018*). Further, research in the Kruger National Park suggests that interspecific interactions may have effects on the distribution of African megafauna, but that this may not be evident at the local scale (*Ryan & Ladau, 2017*). The two-species occupancy results (see Supplementary Material) as well as the high overlap in activity (Fig. 6) suggest that small antelope and managed ungulates occur independently of one another at the local scale. However, given that the small area assessed in this study represents the full area under management (for Postberg at least), larger landscape level analyses are not possible in this context.

## Habitat, forage and cover availability

Ecological theory suggests that for sympatric species to co-exist, subordinate species need to be able to exploit a resource which is not available to dominant species (*Gordon & Illius, 1989*). It is therefore likely that small antelope and managed ungulates have segregated food resources at our study site. This segregation may be due to the difference in body size that dictates forage requirements and forage availability on the vertical and horizontal planes (*Cromsigt & Olff, 2006*). Small antelope would be able to access new growth that is located at a low level or within the shrub itself, whereas the large ungulates are likely utilising the outer and higher parts of the shrub component. Larger ungulates naturally have more access along the horizontal plane as they are not restricted to home ranges (*Jarman, 1974*) and therefore can move greater distances between suitable foraging patches (*Venter et al., 2015*). Additionally, common duiker and steenbok are both known to prefer habitats where shrubs provide adequate cover (*Heydenrych, 1995*).

Vegetation height was found to be more variable in the park, compared to the farm. This was a result of the presence of former cultivated areas (fallow lands). These fallow lands appear to have been maintained by grazing pressure from large-bodied ungulates. Although managed ungulates were detected at a high proportion of sites, they may be transient at many of them and merely passing through. This is supported by the much higher detection of managed ungulates at fallow sites when compared to the more vegetated sites within the park. Research has also suggested that habitat heterogeneity, such as observed in Postberg and Langebaan, may drive movement patterns of larger herbivores as they move between suitable foraging patches (*Venter et al., 2015*).

## Predation risk

Large managed ungulates were less active at night indicating that they were likely meeting their foraging requirements during the day since there is no predation risk for them. In contrast small antelope were more active at night. This likely stems from trade-offs between foraging and vigilance for predators (not required at the current site for larger species) that results in small antelope not meeting their foraging requirements during the day (*Owen-Smith & Goodall, 2014*; *Rowcliffe et al., 2014*). Caracal activity in the park is typical of that observed in other felids (*Ramesh & Downs, 2015*; *Reilly et al., 2017*), being predominantly nocturnal, with a distinct drop in activity between sunrise and midday after which activity picked up again. Small antelope showed a largely inverse, crepuscular pattern. This is indicative of anti-predator behavior portrayed by small antelope. However, there is a degree of temporal overlap between the species (delta ~0.5) and this might suggest that caracals do not present a significant risk as a predator, which is corroborated by previous research which found that caracal diet within the park consists primarily of rodents (84% occurrence in scat), while small antelope were less important with a 6.5% occurrence in scat (*Avenant & Nel, 2002*).

## Caveats

Ecological systems are complex and interactions can be influenced by multiple biotic and abiotic characteristics. The capability of our models was limited due to low variation in the

data, such as detection of small antelope at all sites on the farm, which made elucidating patterns of interest difficult (e.g., *Ramesh & Downs, 2015*) did not even attempt occupancy models where naïve occupancy was 1). Although camera traps are excellent tools for monitoring animal communities, they still have some limitations as they only monitor a fixed point in the landscape. For example, dense vegetation restricts the detection zone, which could have influenced detection of species at vegetated sites. Therefore detection probability is a particularly important consideration when conducting camera surveys, especially surveys that focus on communities rather than an individual species as different species often have specific habitat requirements and are thus detected more frequently in areas where other species might not be detected.

## CONCLUSIONS

Although this study had certain limitations, the results suggest a high level of spatial and temporal overlap between managed ungulates and small antelope. Further, there is an indication that the two groups of species likely occur independently of one another, which is substantiated by the results from the two-species occupancy models (see Supplemental Information). This suggests that competition and facilitation are unlikely drivers of small antelope occurrence, but rather that these sympatric species co-exist due to segregation of food resources. While the low occurrence of small antelope in Postberg might be ascribed to competitive exclusion by managed ungulates, this may rather be a legacy effect of disturbance. Additionally, Postberg is significantly smaller than Langebaan and the farm with significantly steeper slope which may suggest that there is simply less suitable habitat for small antelope within Postberg.

While managed ungulates were detected at a high proportion of sites at Postberg and Langebaan we postulate that occurrence at many of the sites was due to them moving between areas of suitable forage, and not use of the site *per se*. This is supported by the fact that they spent significantly more time on the fallow lands, with lower residence at other sites (Fig. 5). *Radloff (2008)* concluded that eland and bontebok avoided sandstone and limestone Fynbos, and when they did occur there they mainly utilised grassy microhabitats, much like the fallow lands in this study. The managed ungulates are water dependent, whereas the small antelope are not (*Valeix et al., 2009*), and thus would need to travel between water points and forage patches regularly. This made it challenging to assess their influence on small antelope occurrence, relative to movements required to meet their own metabolic needs and resource partitioning.

Overall we found some effects of inter-specific interactions at the local scale but there was a lack of reliable pattern across areas. This is consistent with literature that suggests large-scale ecological trends are difficult to detect at fine scales (*Ryan & Ladau, 2017*; *Wang et al., 2018*). Inability to determine cause and effect has implications for the adaptive management of protected areas since many of South Africa's protected areas are small, fenced and stock ungulate species that have large spatial requirements. The resultant restriction of natural movement patterns within these protected areas, therefore, confounds our ability to detect ecological processes. Considering the financial and human resource

capacity of most small protected areas, this study represents a realistic (if not more so) level of ecological monitoring. This begs the question, can we realistically monitor and understand the impacts of management practices in small protected areas? In addition, replication of historical ecological processes in the small land parcels remaining for protection is uncertain, especially considering species that were transient and whose home ranges exceed the size of the protected area.

### Recommendations

While landowners have considered predator control as a possible means of bolstering small antelope populations, the results provide no clear indication that management intervention is required for the caracal population. Park management have commenced removal of large numbers of game, with the focus on phasing out extralimital species and to reduce the negative impact on vegetation. We expect that this process will allow for vegetation recovery and over time, an associated increase in small antelope. As such, we plan to do follow-up surveys in the future. The current survey is however very important for providing a snapshot of species occupancy and co-occurrence around the time of these removals.

There were many areas where either managed or small ungulates had naïve occupancy close to one. Given the small area of the park sections, there was limited opportunity for additional spatial sampling within Postberg or on a comparable vegetation type within Langebaan. Thus we suggest replicating the experiment in an area that allows greater spatial sampling, though no comparable reserve currently exists in the region. The alternative is to repeat the survey across the whole park (including larger areas and other vegetation types), to better understand requirements for existence of individual species as well as co-existence, which will benefit the management of small protected areas. Should recovery of small antelope to levels observed outside of the park not be observed over time, we recommend deploying GPS collars on managed ungulates and small antelope Langebaan and Postberg to provide a finer scale understanding of spatial and temporal resource partitioning and co-occurrence of these sympatric species.

## ACKNOWLEDGEMENTS

We thank Mila Truter, Trevor Adams, Chanel Williams, Zishan Ebrahim and Toni Dyers for their assistance with field data collection. Thanks to Mila Truter and Alexis Bierman for their assistance with data capture and Chad Cheney for assistance in extracting site information from DEMs. Robert Schlegel (determining independent captures) and Matt Rogan provided advice and inputs into R scripts, for which we are grateful. This work would not have been possible without support from South African National Parks and West Coast National Park management staff. We would also like to thank Christie Rheeder for his support.

### Funding

This work was supported by South African National Parks. The funders had no role in study design, data collection and analysis, decision to publish, or preparation of the manuscript.

### Grant Disclosures

The following grant information was disclosed by the authors:
South African National Parks.

### Competing Interests

The authors declare there are no competing interests.

### Author Contributions

- Deborah Jean Winterton conceived and designed the experiments, performed the experiments, analyzed the data, contributed reagents/materials/analysis tools, prepared figures and/or tables, authored or reviewed drafts of the paper, approved the final draft.
- Nicola J. van Wilgen conceived and designed the experiments, analyzed the data, contributed reagents/materials/analysis tools, prepared figures and/or tables, authored or reviewed drafts of the paper, approved the final draft.
- Jan A. Venter conceived and designed the experiments, contributed reagents/materials/analysis tools, authored or reviewed drafts of the paper, approved the final draft.

### Field Study Permissions

The following information was supplied relating to field study approvals (i.e., approving body and any reference numbers):

Christie Rheeder, the manager of Nortier Research Farm, granted approval to conduct research at the Nortier Research Farm on behalf of the Department of Agriculture (Elsenburg), Western Cape Government.

### Data Availability

The raw data, metadata, and methodology are available in the Supplementary Files.

### Supplemental Information

Supplemental information for this article can be found online at http://dx.doi.org/10.7717/peerj.8184#supplemental-information.

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
