# Peer review of "Investigating the effects of management practice on mammalian co-occurrence along the West Coast of South Africa"

_PeerJ, doi:10.7717/peerj.8184_

## Round 0.1 · original submission · Minor Revisions

This is an important subject and a timely and well developed manuscript, and I think the reviewers have done a great job at helping to improve it. Please address their issues and suggestions and I look forward to seeing an updated submission again soon.

Reviewer 1 ·

Basic reporting

No comment

Experimental design

No comment

Validity of the findings

No comment

Additional comments

The paper shows interesting co-occurrence patterns on how meso-carnivores have an impact on small ungulate interactions under three management scenarios. The paper has important conservation value and management implications for the wildlife species exposed to high livestock pressure. This study is all the more important because wild mammal interactions have been rarely documented in Fynbos biomes, a unique habitat of the Western Cape. The manuscript is written relatively well and analysed with the help of right methods and analytical tools. However, the manuscript lacks some essential information needed to set the basis. The manuscript can be accepted after minor revision. Some of my concerns are highlighted below:
The variables used in the occupancy analysis are important and relevant to the objectives of the study and the fact that the study site is subject to high anthropogenic pressure, however authors must provide justification for each variable and apriori hypothetical expectations on responses of mammalian species at least at the group/community level to covariates in a table which will help readers to understand much better. The study hasn’t accounted seasonal effect on species co-occurrence and distribution as this can play a major role in shaping the species interaction. Authors should highlight about this issue how this seasonal effect will be taken into account as their sampling duration stretches across six months.
Abstract
Line 1 – 12 should be condensed into two simple lines which should explain effect of different level of managements on species distribution and co-occurrence. So that, it will compose entire story of the manuscript in two sentences with wider understanding. Currently, it is written like a popular article. The research question is not well explained that should be stated explicitly in details. I don’t see much results presented in the abstract, authors should highlight and show important results with values.
Not required so delete – “Data were analysed in R, using the unmarked and overlap packages to assess occurrence and temporal activity respectively”.
Delete - “Our inability to detect clear cause and effect has negative implications for adaptive management”.

Introduction
Introduction should explain how management effectiveness can be evaluated on basis of what…………. and How other studies evaluated and its importance.
Authors should highlight in the introduction that past studies done relevant to various land management practices and top predator effect on distribution and co-cooccurrence patterns of various prey species and competitive interaction between herbivore species. What are the research gaps authors have identified from literatures should be mentioned explicitly?
Line 75: “Land use” ? – mention what type of land use/practices.
Line 117: Remove “.”
At the end of the introduction, authors should state how the study is going to benefit the management of the wildlife species as an outcome of the expected results.

Methods
Are you able to measure the height of the vegetation using measure tape?. How was that done and what was the maximum height of the vegetation.
Are there any livelihood dependence pattern of local and their dependence on the park? If yes, then authors could give details of its possible effect better in the park.

Result
The result section requires summary of some additional descriptive information about no. of species recorded across management units, what is the average naïve occupancy range and its difference from the predicted occupancy.
Line 312-313: Add citation for which goodness of fit test was used.
The lines 385-395 must be given a sub-heading “Abundance”
Lines 397-418 must be given a sub-heading “Detection rates”
Authors could also explain about model fit to their data set.
Line 352-353: “This resulted in 1150, 1159 and 1213 trap nights for the Postberg, Langebaan and the farm respectively” should move to the result section.

Line 179-180: Remove – “using an asymptotic test in the cvequality package in R (Marwick & Krishnamoorthy 2018)”.

Discussion
I agree the discussion is written well but evidence-based support from other studies/literature needed for the lines: 463-471. Similar situation might have explained from other studies.
Line 491: Cite the Supplementary Material appropriately
I would strongly suggest authors to provide detailed action-based management suggestions at end of the discussion which would help in improving the Park Management/decision making of the Park Managers.

Figure 1
Camera legend is missing in the map.

·

Basic reporting

No comment

Experimental design

The study investigates species interactions across different management systems in protected areas of South Africa. Authors have applied the most appropriate analytical methods to their data, however the small sample size was a major issue in the study because of which clear conclusions could not be drawn for caracal and ungulate interactions. I also felt that the short duration and sampling effort was relatively low and probably a long-term study and larger sampling coverage could have strengthened their findings.

Validity of the findings

Findings have been stated clearly, however the recommendations lacked more details on how park/habitat management can be improved in these Protected Areas especially for those ungulates with low detections/occupancy. More details on this are needed in the recommendations section.

Additional comments

Overall the study was well executed with valid hypothesis and their objectives were also well framed with appropriate study design. The abstract lacks the statistical outputs/values for the conclusions made. You will see my comments in the annotated pdf attached. Many units are missing in the results section. There is absolute clarity throughout the manuscript and with minor revisions the paper can be accepted for publication.

Reviewer 3 ·

Basic reporting

Well written and seems to cover the literature and background adequately (although I am not very familiar with the literature). Raw data files are shared.

Experimental design

Design is explicitly stated, repeatable and scientifically valid - with three different "treatments" (farm and two reserves), adequate replication with camera traps, equal effort between treatments, and correction of occupancy response variables (of different ungulate groups) for probability of detection.

Validity of the findings

The statistical approach using occupancy models seems to be sound. The study does not seem to have a clear conclusive finding to explain reduced occurrence of small antelopes in the Postberg Reserve. The authors seem to come up with several possible non-exclusive conclusions for this, e.g. predation pressure (some temporal non-overlap with caracals but not conclusive since caracal diet in the area mostly comprises rodents), habitat features such as vegetation height (for duikers), or competition with managed ungulates (also apparently inconclusive since managed ungulates move more between water and food sources while small antelopes are more territorial). The authors acknowledge the caveats and limitations of their approach and suggest a future study using GPS tracking.

Additional comments

No comment

---

## Round 0.2 · accepted · Accept

Your attention to the reviewers comments is appreciated and the final product has certainty improved. Your resubmission looks great. Thank you for considering PeerJ for your manuscript.